# Uncertainties Induced by Processing Parameter Variation in Selective Laser Melting of Ti6Al4V Revealed by In-Situ X-ray Imaging

**DOI:** 10.3390/ma15020530

**Published:** 2022-01-11

**Authors:** Zachary A. Young, Meelap M. Coday, Qilin Guo, Minglei Qu, S. Mohammad H. Hojjatzadeh, Luis I. Escano, Kamel Fezzaa, Tao Sun, Lianyi Chen

**Affiliations:** 1Department of Mechanical and Aerospace Engineering, Missouri University of Science and Technology, Rolla, MO 65409, USA; zay7c4@umsystem.edu (Z.A.Y.); mmcckd@mst.edu (M.M.C.); 2Department of Mechanical Engineering, University of Wisconsin-Madison, Madison, WI 53706, USA; qguo46@wisc.edu (Q.G.); mqu22@wisc.edu (M.Q.); hojjatzadeh@wisc.edu (S.M.H.H.); escanovolque@wisc.edu (L.I.E.); 3Department of Materials Science and Engineering, University of Wisconsin-Madison, Madison, WI 53706, USA; 4X-ray Science Division, Advanced Photon Source, Argonne National Laboratory, Lemont, IL 60439, USA; fezzaa@aps.anl.gov; 5Department of Materials Science and Engineering, University of Virginia, Charlottesville, VA 22904, USA; ts7qw@virginia.edu

**Keywords:** selective laser melting, laser powder bed fusion, additive manufacturing, spatter, melt pool dynamics, quality uncertainty

## Abstract

Selective laser melting (SLM) additive manufacturing (AM) exhibits uncertainties, where variations in build quality are present despite utilizing the same optimized processing parameters. In this work, we identify the sources of uncertainty in SLM process by in-situ characterization of SLM dynamics induced by small variations in processing parameters. We show that variations in the laser beam size, laser power, laser scan speed, and powder layer thickness result in significant variations in the depression zone, melt pool, and spatter behavior. On average, a small deviation of only ~5% from the optimized/reference laser processing parameter resulted in a ~10% or greater change in the depression zone and melt pool geometries. For spatter dynamics, small variation (10 μm, 11%) of the laser beam size could lead to over 40% change in the overall volume of the spatter generated. The responses of the SLM dynamics to small variations of processing parameters revealed in this work are useful for understanding the process uncertainties in the SLM process.

## 1. Introduction

Selective laser melting (SLM, also called laser powder bed fusion) is an additive manufacturing (AM) process that utilizes a high-power density laser to selectively fuse together metallic powders to form three-dimensional objects [1,2,3]. Complex-shaped metal parts for rapid production with high levels of flexibility and customization compared to conventional manufacturing methods is revolutionizing the metal manufacturing industry for aerospace, biomedical, and defense applications [2]. Presently, SLM still faces several challenges: (1) parts printed by the same machine and using the same optimized parameters are not always identical, (2) properties of the printed parts can be difficult to predict, and (3) defect sensitive properties (e.g., fatigue life) of SLM parts are not as good as their wrought counterparts. An understanding of the fundamental mechanisms of SLM and identifying the causes for part quality uncertainty is important for addressing and overcoming the challenges in SLM.

During the SLM process, the interaction between the focused laser beam and the powder bed results in the formation of a cavity due to material vaporization. This vaporization induced cavity is known as the depression zone. Immediately surrounding the depression zone, the powders fuse together to form a localized liquid region known as the melt pool. The melt pool rapidly cools and forms the part. During laser vaporization and melting, liquid droplets can be ejected from the depression zone and melt pool regions, which is called spatter [1].

The four most significant processing parameters [4,5] that can be manipulated to control the SLM process include: (1) laser beam size, (2) laser power, (3) laser scan speed, and (4) powder layer thickness. Each one of these parameters will influence the resulting shape and size of the depression zone, the melt pool, and the spatter behavior. Previous publications show that the processing parameters are critical factors that contribute to the resulting microstructural features and mechanical properties since they influence the thermal history and cooling rates for Ti6Al4V and other AM materials [1,2,6,7,8,9,10,11,12,13,14]. Additional works highlight the importance of the powder layer thickness on the resulting properties of the manufactured part [15,16,17,18]. These works make use of energy density to describe the effects of laser processing parameters on the dynamics of the AM process [8,9,10,11]. Other works use simulations which utilize thermal and fluid flow models to describe heat and mass transfer during the AM process [2,8,9].

Previous works highlight the importance and impact of SLM processing conditions on the finalized part properties. Work by Criales et al. [19] and Ma et al. [20] both utilized finite element modeling to demonstrate that the laser power and laser scan speed change generates significant variations to the peak temperature, melt pool geometry, and properties of AM parts. Experimental work conducted by Roehling et al. [21] demonstrated the impact of beam size and shape on resulting microstructure. Increases in beam size were demonstrated to increase continuity and smoothness of finalized tracks, while beam ellipticity was demonstrated to manipulate microstructure of the AM parts. Work by Han et al. [22] utilized discrete element simulation to analytically demonstrate the change in deposition consistency when varying layer thickness and experimentally validated the resulting microstructure and tensile properties. Results show that the creation of voids and defects within powder layers attributes to increases in porosity and inclusions in as-built parts and decreases in tensile strength.

Previous works have extensively depicted the importance of AM processing conditions on finalized part properties. Specifically, published research has noted the sudden changes in part properties when altering the laser and powder layer conditions. However, previous works have not identified the significance of small changes in processing conditions on the SLM dynamics.

Utilizing in-situ high-speed high-energy high-resolution synchrotron X-ray imaging allows for the dynamics of the SLM during the laser melting process to be observed and analyzed [23,24,25]. Dimensional characteristics of the depression zone and melt pool can be extracted from X-ray images which correspond to the real-time behavior of the material under SLM conditions. The real-time spatter behavior characteristics are also revealed for the SLM process. The real-time analysis of the uncertainty/variation during the SLM process is made possible through in-situ X-ray characterization.

In this work, we investigate the sources of uncertainty in SLM due to deviations from the optimized/reference AM processing parameters for Ti6Al4V through in-situ high-speed X-ray imaging. We reveal the sensitivity of the SLM process to the processing parameters and identify the leading cause of uncertainty by quantifying the percent change in the SLM dynamics (depression zone dynamics, melt pool dynamics, and spatter dynamics) due to the small variations of the four most important processing parameters: (1) laser beam size, (2) laser power, (3) laser scan speed, and (4) powder layer thickness.

## 2. Materials and Method

### 2.1. Materials

Ti6Al4V titanium alloy was used in this study because (1) it has good X-ray transparency, (2) it is the most commonly used titanium alloy [26], and (3) it is of particular interest to the aerospace, biomedical, and defense industry since it is suitable for a wide range of applications due to its high-strength and low-density [1]. The Ti6Al4V powders for testing were purchased from Pyrogenesis Canada Inc. (Montreal, QC, Canada). The Ti6Al4V metal substrate was purchased from McMaster (Elmhurst, IL, USA). The powder morphology and size distribution are shown in Figure 1.

The chemical compositions of the feedstock powders are shown in Table 1. The two powders have slightly different oxygen and nitrogen. Testing of laser processing parameters was conducted using 15–25 μm powder. Testing of powder layer thickness was conducted using both 15–25 μm and 38–45 μm powders.

### 2.2. In Situ High-Speed Synchrotron X-ray Imaging Experiment

Figure 2 illustrates the schematic of the in-situ high-speed X-ray imaging system. A high-flux synchrotron X-ray with a first harmonic energy of 24 keV and an energy bandwidth of 5~7% was utilized to reveal the dynamics of the SLM process (Beamline 32-ID-B, Advanced Photon Source, Argonne National Laboratory). The transmitted X-ray signal is captured by a scintillator (LuAG:Ce, 100 µm thickness), where the signal is converted into visible light and recorded by a high-speed camera (Photron FastCam SA-Z, Tokyo, Japan) [24]. A frame rate of 50 kHz and a camera exposure time of 1 µs was used to capture the laser melting process. The field of view for the X-ray is 768-pixel × 512-pixel with a resolution of ~2 µm per pixel. The laser scan length is 2.5 mm. The typical sample assembly which is composed of a miniature Ti6Al4V metal substrate with a thickness of 0.40 mm, a height of 2.95 mm, and a powder bed layer thickness of 100 µm is sandwiched between two pieces of glassy carbon plates, which is transparent to the incident X-ray beam. For more details about the in-situ X-ray imaging experiment, refer to previous publications [24,25,27]. Image processing was done using ImageJ to adjust the brightness and contrast of the images to enhance the visibility of the melt pool and depression zone boundaries [28].

### 2.3. Characterization and Quantification of the Sources of Uncertainty in Selective Laser Melting

Figure 3a shows the major features of a substrate during laser scanning and their locations. Figure 3b highlights the various dynamics of the SLM process which are of interest: (1) 2D projection of the depression zone geometry, (2) 2D projection of the melt pool geometry, and (3) spatter behavior. Figure 3c is the 2D projected image of the melt pool boundary, revealing the melt pool depth and length. Figure 3d is an optical image of the top surface of the metal substrate after laser scanning which is used to measure the width of the melt pool after excess powder has been removed. Figure 3e shows the 2D projection of the depression zone geometry, revealing the depression zone depth and width. Figure 3f shows the spatter dynamics. The spatter diameter and volume are measured assuming a spherical spatter geometry. The spatter ejection speed and angle are measured relative to the horizontal location of the top surface of the metal substrate and the laser scanning direction.

Tracking of SLM dynamics is conducted by manual image processing and image analysis. Identification and tracking of spatter, depression zone, and melt pool dynamics are difficult to automate due to changes in the intensity during X-ray scanning. For manual analysis, all measurements are accurate to 1 pixel within the frame. Depression zone dynamics are analyzed at every other frame when the entire region is visible within the field of view. The width is determined to be the region at the top of the substrate where the edges of the depression zone are vaporized due to laser heating. The depth of the depression zone is defined as the distance from the top of the substrate to the lowest point in the depression zone (deepest vaporized region within the substrate). For melt pool analysis, three dimensions are analyzed and measured: the length, depth, and width are determined at every other frame (~40 μs). The length of the melt pool is taken as the farthest liquid region ahead of the depression, to the tail or farthest region where liquid is present (edge of tail). The depth is defined as the vertical distance from the lowest melted region within the substrate to the top of melted region within the powder bed. The location of the top of the melt pool within the powder bed can be identified due to the visible, quantifiable change in intensity from the X-ray image. The width of the melt pool is the only ex-situ analysis conducted within this work. The width of the melt pool is the distance between the edges of the laser melted zone of the single line laser scanning after solidification, measured from the top surface of the sample by optical microscope, as shown in Figure 3d.

Analysis of the spatter requires frame by frame tracking to determine the spatter dynamics. Spatter analysis was solely done for liquid spatter ejection due to the significance of liquid spatter on finalized part properties highlighted in work by Ali et al. [29]. Four main spatter features are tracked: spatter ejection angle, spatter speed, spatter diameter, and spatter volume. The spatter ejection angle is defined as the angle of the spatter ejection relative to the laser scan direction, as indicated in Figure 3f. The spatter speed is the moving speed of the spatter calculated from the in-situ X-ray images. The spatter velocity projected on the 2D imaging plane is calculated using the following equation:(1)$$V_{spatter}=\frac{\sqrt{{\left(Y_{2}-Y_{1}\right)}^{2}+{\left(X_{2}-X_{1}\right)}^{2}}}{t_{2}-t_{1}}$$
where $Y_{2}$, $X_{2}$ and $Y_{1}$ and $X_{1}$ are the cartesian coordinates of the spatter at moment *t*_2_ and *t*_1_, respectively. The average of the vertical diameter and horizontal diameter is used as spatter diameter:(2)$$d=\frac{d_{vertical}+d_{horizontal}}{2}$$
where $d_{vertical}$ and $d_{horizontal}$ are vertical and horizontal diameter of the spatter, respectively.

For simplicity, a spherical geometry is assumed to calculate the volume of spatter produced.
(3)$$v_{spatter}=\sum_{i=1}^{n}\left(\left(\frac{\pi }{6}\right)d_{n}^{3}\right)$$
where n is the number of spatters, *d* is the spatter diameter.

Table 2 summarizes the processing parameters studied in this work. The optimized/reference processing parameters needed for Ti6Al4V under SLM conditions are indicated by the 0% change. Table 2 also details the variations in the processing parameters from the optimized/reference parameters that were studied, along with the percent change in the parameters relative to the optimized/reference parameter (0% change). In this work, we measure and quantify the dynamics of Ti6Al4V under SLM conditions. The characteristic dimensions and quantities of the SLM dynamics are measured for each of the processing parameter conditions. The average value and standard deviation of the characteristic dimensions and quantities are determined. 

### 2.4. Selection of Processing Parameters to Vary

The four most important processing parameters (laser beam size, laser power, scan speed, and powder layer thickness) that influence the melting dynamics during single track laser melting were selected to vary for this study. These four processing parameters can have uncertainty during SLM process due to machine condition drift, part design, part size, or powder spreading uncertainty. Figure 4 depicts the potential causes of the variation of the four processing parameters in commercial AM machine.

The fluctuation in laser beam size may happen in large scale powder bed manufacturing machine due to large build platforms. The laser spot at regions far from the laser origin may have a larger spot size due to the greater distance and angle as depicted in Figure 4a. Work by Ayoola et al. [30] demonstrates this phenomenon of beam size change in conduction mold welding when operating near build platform edges. Laser power and scan speed may also fluctuate within a single AM system despite pre-set operating conditions due to machine parameter drift as seen in Figure 4b,c. These drifts are depicted in work by Moges et al. [31] and highlighted in the work by Lopez et al. [32]. Scan speed variation is influenced primarily by the system scan strategy. Introducing scan strategies with laser start, stop, and directional changes during the active laser scanning generate regions with sudden variations in operating laser scan speeds due to acceleration/deceleration of the scanning mirror. Work by Jia et al. [33] demonstrates unique scan strategies that implement directional changes during laser scanning, causing sudden acceleration or deceleration at start and stop locations. Primarily, this problem has been remedied by increased understanding and g-code manipulation to maintain consistent scan speed velocities but may still occur in certain scanning strategies. Layer thickness fluctuation depicted in Figure 4d is largely driven by the inconsistent flowability of commercial powder leading to reductions in build height and bed density [34]. Work by Jacob et al. [35] demonstrated a measurement procedure to capture the powder bed density and discovered a ~20% fluctuation in the powder bed density along the spreading area. Work by Dowling et al. [36] studied powder bed fluctuations involving powder size, size distribution, and density; results highlight the uncertainty in AM processing and effects on final part properties. It is important to identify and understand these variations in SLM process.

## 3. Results and Discussion

### 3.1. Laser Beam Size Variation

Figure 5 shows the change in SLM dynamics induced by altering the laser beam size during SLM of Ti6Al4V. Figure 5a,b shows the changes in the depression zone depth and width due to the change in the laser beam size. Figure 5c–e depicts the change in melt pool length, depth, and width due to the change in laser beam size. Figure 5f–i demonstrates the changes in the spatter dynamics due to change in beam size. Noticeable trends are highlighted and marked in red. Testing was all conducted with 15–25 µm plasma atomized powder.

The results in Figure 5a indicate that an increase in the laser beam size will result in a decrease in the depression zone depth. A liner trend between the laser beam size and the depression zone depth with a slope of −2.17 was observed. No distinct trend was observed for the change in the depression zone width as shown in Figure 5b. Within the range of laser beam sizes studied, the standard deviation of the depression zone width at each of the seven beam sizes increases significantly once increased at and above 95 µm. The standard deviation remains similar for depression zone depth across the various laser beam sizes. The standard deviation indicates the stability of the depression zone geometry at each laser beam size. A large standard deviation means that there is an instability in the depression zone geometry, resulting in fluctuations during laser scanning. Conversely, a small standard deviation indicates stability in the depression zone. The increasing laser beam size greater than 92 µm generated substantial fluctuation to multiple SLM dynamics marking an instability being present in the depression zone. Previous work by Suzuki et al. [37] demonstrated the effect of alteration of deposited energy density on respective material properties. Beam size results were experimentally demonstrated in our work, showing a decrease in the laser beam size can lead to a larger keyhole depth, which is consistent with the previous work.

Figure 5c–e shows that an increase in the laser beam size will result in a decreasing trend in the melt pool dimensions. A linear trend between the laser beam size and the melt pool dimensions with a slope of −2.334 and −2.077 was observed for the melt pool depth and width, respectively. An increase in the laser beam size increases the size of laser material interaction area, reducing the input energy density. This decrease in the energy density results in changes in the melt pool geometry and is reflected in the data. An increase in the laser beam size leads to a decrease in the melt pool depth and width. A trend in the melt pool length due to a change in the laser beam size was not observed. However, the standard deviation of the melt pool length at the larger beam sizes (95 and 100 µm) is significantly larger than the standard deviations at the smaller laser beam sizes (<95 µm). As the beam size increases and the energy density decreases, there is not sufficient energy to maintain a consistent melt pool shape, resulting in a fragmentation or fluctuation of the melt pool length.

The results in Figure 5f–i show the effects of beam size variation on the overall spatter dynamics. Within the range of testing, the spatter average diameter, maximum diameter, and direction was not significantly or noticeably influenced by the change in the laser beam size. The increase in beam size, however, led to an increase in the total spatter volume. The change in the laser beam size caused increases or reduction of the spatter production by 47 and 70%, respectively. An increased beam size increases the heat affected zone of the laser with a reduced laser intensity. The decreased laser intensity in the expanded region generated an increased zone for the liquid spatter to form and escape without proper substrate fusion.

As presented above, the small variations in the laser beam size led to significant influence on the overall SLM dynamics. The detailed percent changes are summarized in Table 3.

### 3.2. Laser Power Variation

Figure 6 shows the change in SLM dynamics induced by altering the laser power during SLM of Ti6Al4V. Figure 6a,b shows the changes in the depression zone depth and width due to the change in the laser power. Figure 6c–e depicts the change in melt pool length, depth, and width due to the change in laser power. Figure 6f–i demonstrates the changes in the spatter dynamics due to change in laser power. Testing was all conducted with 15–25 µm plasma atomized powder.

For both the depression zone depth and width in Figure 6a,b, an overall increasing trend was observed due to an increase in the laser power. A linear trend between the laser power and the depression zone dimension with a slope of 0.6451 $\frac{\mu m}{W}$ and 0.4620 $\frac{\mu m}{W}$ was observed for the depression zone depth and width, respectively. Similar trends were obtained by Yin et al. [38] while utilizing in-situ optical imaging techniques at a much wider laser power range (750–1550 W). Our work expands on the work of Yin et al. demonstrating a minute increase to the laser power still leads to a larger depression zone. For both the depression zone depth and width, the standard deviation at each laser power increment were similar.

Figure 6c–e shows that an increase in the laser power leads to an increase in the melt pool size. A linear trend with a slope of 0.4857 $\frac{\mu m}{W}$, 4.103 $\frac{\mu m}{W}$, and 0.2828 $\frac{\mu m}{W}$ were observed for the melt pool depth, length, and width, respectively. A higher laser power allows for the formation of melt pools that are deeper, wider, and longer during the SLM process.

Figure 6f–i depicts the spatter dynamics due to variations in the laser power during laser scanning. The spatter average diameter, direction, speed, and volume were determined. Within the range of testing, no significant trends were observed for the spatter dynamics due to the small alterations in the laser power. For the laser power, the speed and diameter of the spatter had the greatest fluctuations for all testing conditions.

The detailed percent changes in the SLM dynamics induced by variations in laser power are summarized in Table 4.

### 3.3. Laser Scan Speed Variation

Figure 7 shows the change in SLM dynamics induced by altering the laser scan speed during SLM of Ti6Al4V. Figure 7a,b shows the changes in the depression zone depth and width due to the change in the laser power. Figure 7c–e depicts the change in melt pool length, depth, and width due to the change in laser scan speed. Figure 7f–i demonstrates changes in the spatter dynamics due to the change in laser scan speed. Testing was all conducted with 15–25 µm plasma atomized powder.

Figure 7a,b shows that an overall decreasing trend was observed for the depression zone depth and width due to an increase in the laser scan speed. For the depth and width dimensions, a linear trend between the laser scan speed and the depression zone dynamics with a slope of −186.1 $\frac{\mu m\cdot s}{m}$ and −87.32 $\frac{\mu m\cdot s}{m}$ was observed, respectively. An increase in the laser scan speed reduces the input energy density to the powder and substrate materials as discussed by Boswell et al. [39]. This phenomenon is demonstrated in a previous work by Cunningham et al. [40] over a wide range of processing speeds for bare plate testing. Cunningham et al. utilized high speed X-ray imaging to capture the location and penetration depth of the laser; variations in scan velocities from 0.4 to 1.2 m/s lead to significant changes to the size and shape of the depression zone. The standard deviations of the depression zone depth and width at different laser scan speeds were similar, meaning that the fluctuation from the average depression zone depth and width value at each laser scan speed was not significantly affected by the change in laser scan speed.

Figure 7c–e shows that an increase in the laser scan speed leads to a decrease in the melt pool size. A decreasing trend with a slope of −118.1 $\frac{\mu m\cdot s}{m}$, −2121 $\frac{\mu m\cdot s}{m}$, and −285 $\frac{\mu m\cdot s}{m}$ was observed for the melt pool depth, length, and width, respectively. As the laser scan speed increases, the laser interaction time decreases and leads to a decrease in the energy deposition. This decrease in the energy deposition causes less material being melted or fused together, resulting in a smaller melt pool, which is consistent with the previous research results on scan speed variation in a large range of 500 to 1200 mm/s [14]. In terms of the melt pool dynamics fluctuations, no trend was observed in our study as indicated by standard deviation (error bar) in Figure 7c–e.

Figure 7f–i depicts the effect of laser scan speed on the spatter dynamics. The spatter’s direction, speed, average diameter, and volume were quantified. No clear trend was observed.

The detailed percent changes in the SLM dynamics are categorized and given in Table 5.

### 3.4. Layer Thickness Variation

Figure 8 shows the change in SLM dynamics induced by altering the layer thickness during SLM of Ti6Al4V. Figure 8a,b shows the changes in the depression zone depth and width due to the change in the layer thickness. Figure 8c–f demonstrates changes in the spatter dynamics due to the change in layer thickness. Testing was conducted with 15–25 µm and 38–45 µm plasma atomized powders.

Figure 8a,b depicts the depression zone dynamics change due to the alteration in the layer thickness. We observed large increases in both the depression zone depth and width due to a reduction in the layer thickness. The 15–25 μm powder experienced an increase of 56 and 33% for the depression zone depth and width, respectively, due to the 50 μm layer thickness reduction; 38–45 μm powder experienced an increase of 33 and 55% for the depth and width, respectively. The standard deviation of the depression depth and width was not significantly affected by the layer thickness change.

Figure 8c–f demonstrates the effect layer thickness on the spatter dynamics. For the effect of layer thickness on spatter size, no trend was observed. However, the decrease in the layer thickness increased the average spatter angle for both powder sizes. The total volume of spatter was increased as the layer thickness increases from 50 µm to 100 µm for 15–25 μm powder. We also observed that changing the powder size from 15–25 μm to 38–45 μm results in the increase of total spatter volume produced during laser scanning, which can be attributed to the effect of powder size on inter-particle laser reflection/absorption as discussed in the work by Zhang et al. [41]. 

Altering the layer thickness had a substantial change on the overall SLM dynamics. The detailed percent changes in the SLM dynamics are categorized and given in Table 6.

### 3.5. Sensitivity Analysis of Processing Conditions

Testing sensitivity of process dynamics to processing condition variation is important for understanding process repeatability in SLM. Work by Yadav et al. [42] and Dowling et al. [36] outlined the importance and necessity for limiting the causes of uncertainty to tackle the quality uncertainty challenge in AM. Works by Kusuma et al. [43], Nguyen et al. [18], and Hanzl et al. [44] discussed in detail the effects of processing parameter variation on finalized part properties. Our work induced small parameter variations to processing conditions, revealing the sensitivity of the SLM dynamics to intrinsic process variations.

Layer thickness variation affects the SLM dynamics by the same amount as the percent variation of layer thickness. The variations of the other three processing parameters (laser power, scan speed, and beam size) lead to the percentage changes of SLM dynamics more than the percentage changes of the processing parameters. Laser beam size change of no greater than 11% resulted in changes of the depression zone and melt pool dynamics by 26%. The laser power change of only 5% caused changes of the depression zone and melt pool dynamics by 12%. The scan speed variation of 5% generated up to 16% of depression zone and melt pool fluctuations.

Current AM systems operating under optimized processing conditions may still encounter intrinsic processing parameter drifts, causing significant changes to the underlying SLM dynamics, consequently leading to part reproducibility issues. Understanding the impact of parameter variations on process dynamics is critical for the development of control techniques to achieving reliable metal AM.

## 4. Conclusions

In this work, the sources of uncertainty in the SLM process, for Ti6Al4V, due to variations in the processing parameters (laser power, scan speed, beam size, and layer thickness) were observed, analyzed, and characterized by in-situ X-ray imaging. The major conclusions are summarized below.
Small changes in laser beam size (<12%) from optimized/reference processing parameter produce significant changes on the SLM depression zone and melt pool dynamics. Laser beam size also strongly influences the production of liquid spatter, causing changes up to 70% in the total spatter volume production, indicating that the laser beam size is the most influential processing parameter for spatter control.Laser power fluctuations of 5% generated changes greater than twice the change in the laser power. Specifically, the laser power fluctuation directly affects depression zone and melt pool dynamics, changing the melting region and process stability.Laser scan speed generated the most substantial impact on the depression zone and melt pool dynamics. Scan speed fluctuations of 5% caused up to 15% changes in the depression zone and melt pool dynamics. The control of the laser scan speed during AM processing is vital for mitigating uncertainty.Powder layer thickness fluctuations demonstrated a roughly equivalent effect to fluctuations to the SLM dynamics. The layer thickness primarily controls the layer-by-layer deposition height and did not statistically have an unexpected change to the system dynamics. The increase in powder size, however, showed a sudden increase in the liquid spatter volume production.

## Figures and Tables

**Figure 1 materials-15-00530-f001:**
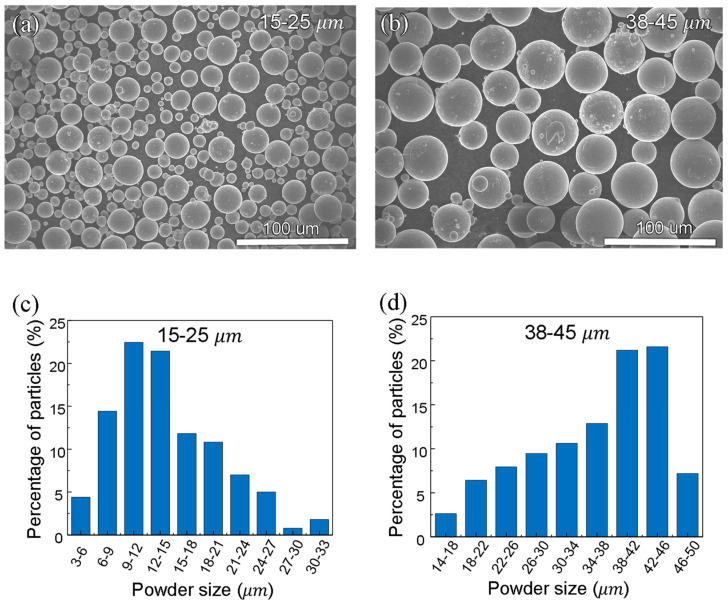
Morphology and size distribution of the feedstock powders. (**a**,**b**) SEM images of Pyrogenesis 15–25 μm powders (**a**) and Pyrogenesis 38–45 μm powders (**b**). (**c**,**d**) Particle size distribution of Pyrogenesis 15–25 μm powders (**c**) and Pyrogenesis 38–45 μm powders (**d**). The percentage is number percentage.

**Figure 2 materials-15-00530-f002:**
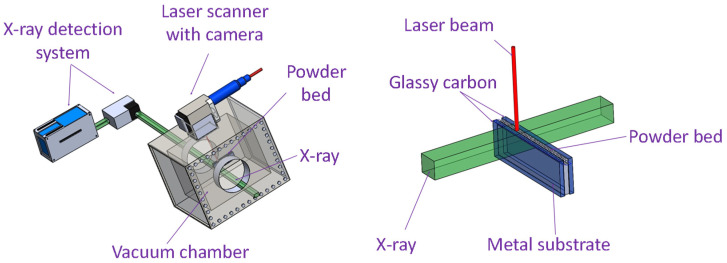
Schematic of the in-situ high-speed synchrotron X-ray imaging system and the sample assembly. The X-ray passes through the sample and is detected by the X-ray detection system. A visible light camera is used to ensure proper laser-sample alignment. Two glassy carbon walls are used to hold the metal substrate and the powder bed while ensuring X-ray transparency along the X-ray beam path. More details regarding in-situ high-speed synchrotron X-ray imaging are in reference [27].

**Figure 3 materials-15-00530-f003:**
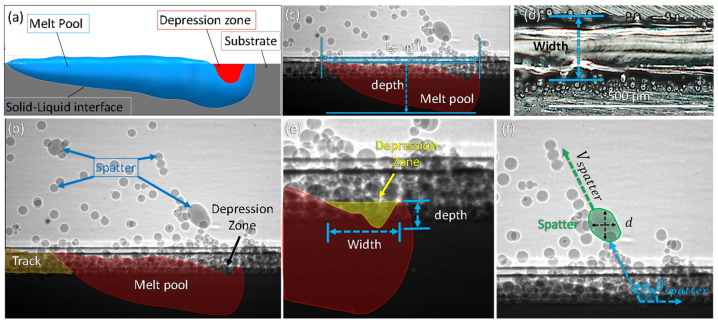
Dynamics of SLM. (**a**) Schematic outlining dynamics during laser scanning. (**b**) The key features observed by in-situ X-ray imaging. (**c**) Typical melt pool length and depth dimensions. (**d**) Typical optical image of the melt pool width measured after laser melting and removal of excess powder. (**e**) Typical depression zone depth and width dimensions. (**f**) Typical X-ray image showing spatter dynamics. The spatter diameter, d, spatter ejection angle, θ, and spatter speed, V_spatter_, are indicated in the image.

**Figure 4 materials-15-00530-f004:**
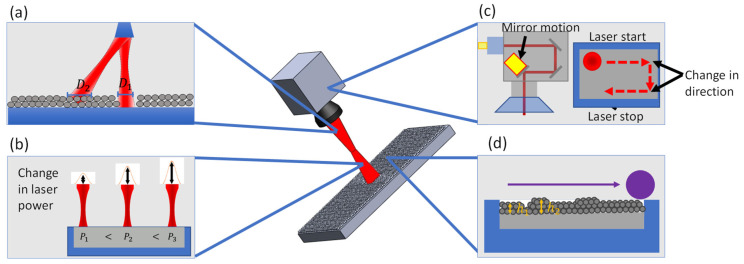
Potential causes of processing parameter variation. (**a**) Variation of beam size in large build due to increased distance on build edge locations. (**b**) Variation of laser power due to intrinsic laser power drift. (**c**) Variation of laser scan speed due to intrinsic drift, turning, starting, and stopping. (**d**) Variation of powder layer thickness due to inconsistent powder flowability and spreading.

**Figure 5 materials-15-00530-f005:**
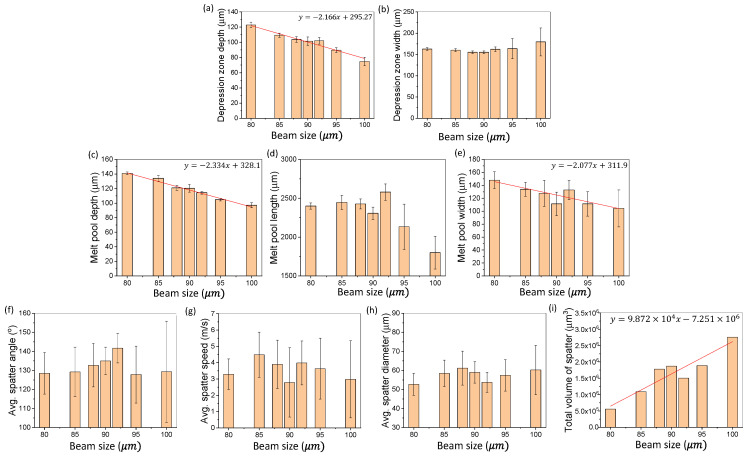
Effects of laser beam size variation on SLM dynamics. (**a**,**b**) Changes in the depression zone depth and width due to variation in the laser beam size. (**c**–**e**) Changes of the melt pool depth, length, and width due to variations in laser beam size. (**f**–**i**) Changes of the spatter average direction, speed, diameter, and total volume due to variation in the laser beam size. All testing is conducted using 15–25 µm, plasma atomized Ti6Al4V powder.

**Figure 6 materials-15-00530-f006:**
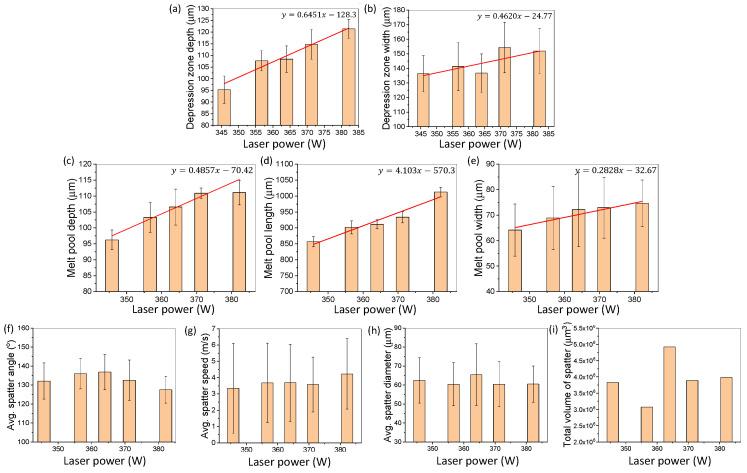
Effects of laser power variation on SLM dynamics for Ti6Al4V. (**a**,**b**) Changes in the depression zone depth and width due to variation in the laser power. (**c**–**e**) Changes of the melt pool depth, length, and width due to variations in laser power. (**f**–**i**) Changes of the spatter average direction, speed, diameter, and total volume due to variation in the laser power. All testing is conducted using 15–25 µm, plasma atomized Ti6Al4V powder.

**Figure 7 materials-15-00530-f007:**
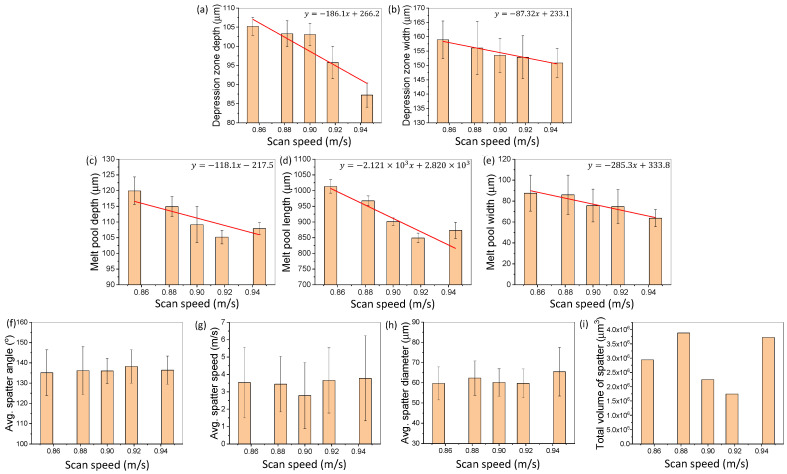
Effects of laser scan speed variation on SLM dynamics. (**a**,**b**) Changes in the depression zone depth and width due to variation in the laser scan speed. (**c**–**e**) Changes of the melt pool depth, length, and width due to variations in laser scan speed. (**f**–**i**) Changes of the spatter average direction, speed, diameter, and total volume due to variation in the laser scan speed. All testing is conducted using 15–25 µm, plasma atomized Ti6Al4V powder.

**Figure 8 materials-15-00530-f008:**
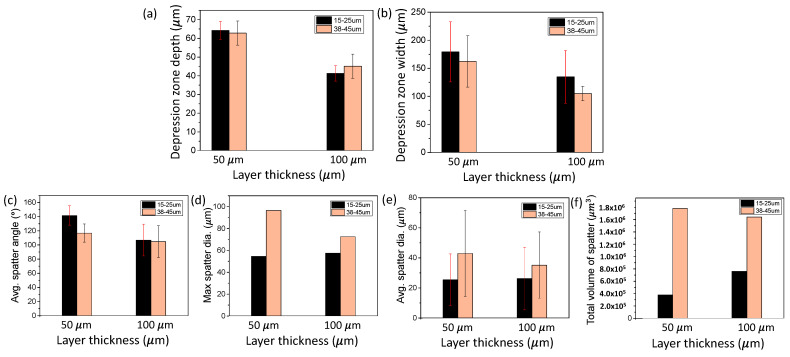
Effects of layer thickness variation on SLM dynamics. (**a**,**b**) Changes in the depression zone depth and width due to variation in the layer thickness. (**c**–**f**) Changes in the spatter average direction, maximum diameter, average diameter, and total volume due to variation in the layer thickness. Testing is conducted using 15–25 µm and 38–45 µm, plasma atomized Ti6Al4V powders.

**Table 1 materials-15-00530-t001:** Chemical composition of Ti6Al4V by wt.%.

Element	15–25 μm	38–45 μm
Titanium	Balance	Balance
Aluminum	5.5	5.5
Vanadium	3.5–4.5	3.5–4.5
Carbon	<0.08	<0.08
Oxygen	0.16	0.12
Nitrogen	0.02	0.01
Hydrogen	<0.015	<0.015
Iron	<0.40	<0.40
Other total, max	0.40	0.40

**Table 2 materials-15-00530-t002:** Experimental processing parameters for SLM of Ti6Al4V. Powder bed plane is indicated by the distance from the focal plane of the laser (negative sign indicates that the powder bed plane is below the focal plane of the laser).

Variation in laser beam size							
Beam size, *D* (μm)	Powder bed plane, *d* (mm)	Power, *P* (W)	Scan speed, *V* (m/s)	Scan length, *l* (mm)	Powder layer thickness, *t* (μm)		Beam size change (%)
80	−2	364	0.9	3	100		−11
85	−2.25						−6
88	−2.4						−2
90	−2.5						0
92	−2.6						2
95	−2.75						6
100	−3						11
Variation in laser power							
Beam size, *D* (μm)	Powder bed plane, *d* (mm)	Power, *P* (W)	Scan speed, *V* (m/s)	Scan length, *l* (mm)	Powder layer thickness, *t* (μm)		Power change (%)
90	−2.5	346	0.9	3	100		−5
		357					−2
		364					0
		371					2
		382					5
Variation in laser scan speed							
Beam size, *D* (μm)	Powder bed plane, *d* (mm)	Power, *P* (W)	Scan speed, *V* (m/s)	Scan length, *l* (mm)	Powder layer thickness, *t* (μm)		Scan speed change (%)
90	−2.5	364	0.855	3	100		−5
			0.882				−2
			0.9				0
			0.918				2
			0.945				5
Variation in layer thickness							
Beam size, *D* (μm)	Powder bed plane, *d* (mm)	Power, *P* (W)	Scan speed, *V* (m/s)	Powder size (μm)	Powder layer thickness, *t* (μm)	Scan length, *l* (mm)	Powder layer thickness change (μm)
90	−2.5	260	1.0	15–25	50	3	50
				15–25	100		0
				38–45	50		−50
				38–45	100		0

**Table 3 materials-15-00530-t003:** Percent change in SLM dynamics induced by variations in laser beam size during laser scanning.

Beam size: depression zone dynamics					
Beam size, *D* (μm)	Beam size change (%)		Depth change (%)		Width change (%)
80	−11		21		5
85	−6		8		3
88	−2		2		0
90	0		0		0
92	2		0		5
95	6		−12		6
100	11		−26		16
Beam size: melt pool dynamics					
Beam size, *D* (μm)	Beam size change (%)	Depth change (%)		Length change (%)	Width change (%)
80	−11	17		4	33
85	−6	11		6	20
88	−2	1		5	15
90	0	0		0	0
92	2	−5		12	19
95	6	−13		−8	0
100	11	−19		−22	−6
Beam size: spatter dynamics					
Beam size, *D* (μm)	Beam size change (%)	Ejection angle change (%)	Ejection speed change (%)	Spatter diameter change (%)	Spatter volume change (%)
80	−11	−5	−17	−11	−70
85	−6	−4	13	−1	−22
88	−2	−2	−2	4	−5
90	0	0	0	0	0
92	2	5	−9	−9	−20
95	6	−5	−25	−3	1
100	11	−4	−30	2	47

**Table 4 materials-15-00530-t004:** Percent change in SLM dynamics induced by variations in laser power during scanning.

Laser power: depression zone dynamics					
Laser power, P (W)	Power change (%)		Depth change (%)		Width change (%)
346	−5		−12		0
357	−2		−1		3
364	0		0		0
371	2		6		13
382	5		12		11
Laser power: melt pool dynamics					
Laser power, P (W)	Power change (%)	Depth change (%)		Length change (%)	Width change (%)
346	−5	−10		−6	−11
357	−2	−3		−1	−5
364	0	0		0	0
371	2	−4		2	1
382	5	4		11	3
Laser power: spatter dynamics					
Laser power, P (W)	Power change (%)	Ejection angle change (%)	Ejection speed change (%)	Spatter diameter change (%)	Spatter volume change (%)
346	−5	−16	−9	−16	−22
357	−2	−16	0	−16	−38
364	0	0	0	0	0
371	2	−8	−3	−8	−21
382	5	−13	15	−13	−19

**Table 5 materials-15-00530-t005:** Percent change in SLM dynamics induced by variations in the laser scan speed.

Laser scan speed: depression zone dynamics					
Laser scan speed, V (m/s)	Scan speed change (%)		Depth change (%)		Width change (%)
0.855	−5		2		4
0.882	−2		0		2
0.9	0		0		0
0.918	2		−7		0
0.945	5		−15		−2
Laser scan speed: melt pool dynamics					
Laser scan speed, V (m/s)	Scan speed change (%)	Depth change (%)		Length change (%)	Width change (%)
0.855	−5	10		12	−16
0.882	−2	5		7	14
0.9	0	0		0	0
0.918	2	−4		−6	−1
0.945	5	−1		−3	−16
Laser scan speed: spatter dynamics					
Laser scan speed, V (m/s)	Scan speed change (%)	Ejection angle change (%)	Ejection speed change (%)	Spatter diameter change (%)	Spatter volume change (%)
0.855	−5	−0.7	27	−1	31
0.882	−2	0.1	24	4	73
0.9	0	0	0	0	0
0.918	2	1.6	31	−1	−22
0.945	5	0.3	36	−9	66

**Table 6 materials-15-00530-t006:** Percent change in SLM dynamics induced by variations in the layer thickness.

Layer thickness: depression zone dynamics					
Powder size/layer thickness (µm/µm)	Layer thickness change (%)		Depth change (%)		Width change (%)
15–25/100	0		0		0
15–25/50	50		56		33
38–45/100	0		0		0
38–45/50	50		39		55
Layer thickness: spatter dynamics					
Powder size/layer thickness (µm/µm)	Layer thickness change (%)	Ejection angle change (%)	Ejection speed change (%)	Spatter diameter change (%)	Spatter volume change (%)
15–25/100	0	0	0	0	0
15–25/50	50	33	−5	−3	−34
38–45/100	0	0	0	0	0
38–45/50	50	11	33	22	8

## Data Availability

Data sharing is not applicable.

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
