# Peer review of "Uncertainties Induced by Processing Parameter Variation in Selective Laser Melting of Ti6Al4V Revealed by In-Situ X-ray Imaging"

_materials, 2022, doi:10.3390/ma15020530_

Round 1
Reviewer 1 Report
This manuscript proposes, “Uncertainties induced by processing parameter variation in selective laser melting of Ti6Al4V revealed by in-situ x-3 ray imaging”. The topic is interesting with the contents to be proposed to the readers of Materials Journal. However, the manuscript should be improved to be read with pleasure in terms of bibliographic updates, grammar corrections and content deepening. Overall, I think that this manuscript could be accepted if the Authors will be able to take into account the following major revisions
- Introduction section needs extensive revision
- Plagiarism is very high (more than 45%)
- I didn’t found any past work of researchers in literature related to topic. Authors should include it in introduction section and write research gap
- Justification about selection of processing parameters needs more attention
- Add chemical composition of selected material (Ti6Al4V)
- In results and discussion section, there is not a single reference used to explain the results.
- Error bar in Figure 4 (i) is missing
- Authors may write the conclusion section point wise
Reviewer 2 Report
It’s very significant to reveal the sensitivity of the SLM process to the processing parameters and identify the leading cause of uncertainty by quantifying the percent change in the SLM dynamics. In the manuscript, the depression zone depth and width, the melt pool depth, length, and width, spatter average direction, speed, diameter, and total volume under various laser beam sizes, laser powers, laser scan speeds, and powder bed thickness were measured by In-situ high-speed synchrotron x-ray imaging experiment. But, the manuscript seems to be more like an engineering report than a sci-tech paper. I cannot find any breakthroughs in scientific theory and technology. The methods of measurements mathematical statistics on the SLM dynamics are missing. It is significant to evaluate the data reliability. Moreover, the writing style of manuscript is not an effective and refined way of expression. There are so many repetitive and redundant expressions in the manuscript. It is should be improved to refine the written expression. Therefore, I think it is not suitable for publication in the present form.
Round 2
Reviewer 1 Report
All the questions raised in the previous round of review have been addressed by the authors. In my opinion, manuscript can now be accepted after incorporating the following comment.
1) Reference can be added immediately after authors name in (e.g. Line 75: Criales et al. [19] and Ma et al. [20] like this) instead of adding the reference after the completion of sentence. Revise this throughout the manuscript.
Reviewer 2 Report
I am satisfied with the response.
Author Response
Dear Reviewer,
Thank you for pointing out the necessary improvements to demonstrate the scientific discovery of the work. These changes significantly improved the quality of this work. Thank you as well for the approval of the changes to the revised manuscript.
Kindest Regards,
Zachary Young